# OpenReview forum: "Birkhoff-Exact Hyper-Connections: Exact Spectral Stability for Deep Residual Networks"
_ICLR.cc/2026/Workshop/Sci4DL — Sci4DL 2026_

### Official Review · Reviewer_FvzV · 2026-02-14

**Fit:** 3
**Significance:** 2
**Confidence:** 2

**Summary:**

The paper is mainly based on the viewpoint: **exact—not approximate—spectral stability is necessary for extreme-depth training**. Starting  from this, the paper proposes Birkhoff-Exact Hyper-Connections (BE-HC): an exactly doubly stochastic mixing operator constructed as a convex combination of *fix-size* permutation matrices via the Birkhoff–von Neumann theorem. It provides theoretical guarantees on spectral stability and mean conservation, and reports stable 1000-layer training on CIFAR-100 plus scalability/efficiency/quantization experiments. Furthermore, the paper investigates the impact of different basis choices, demonstrating that the appropriate configuration aligns with the inductive biases inherent in vision tasks. Other related perspectives, such as interpretability, are also studied.

**Strengths:**

**Significance of the raised question**: Good and Important.

**Evidence supporting the viewpoint answering the raised question**: Medium, not fully convincing.

**Soundness of the methodology**: Good, but can be improved.

**Strengths:**
1. The proposed architecture is *simple yet effective* to guarantee exact doubly stochasticity and avoid iterative normalization error accumulation.
2. The paper also gives a sequential way to implement the method efficiently with stated complexity O(Knd) and reduced memory from O(Kn²) to O(Kn). Moreover, as mentioned in the appendix, they also propose the vectorized implementation, which further exploits the GPU, achieving 30-50% speed up over the sequential implementation.
3. BE-HC has theoretical guarantees on spectral stability and mean conservation, serving as the foundation of stable training.
4. Several experiments across different areas are done to validate the BE-HC architecture.

**Suggestions:**

**Major Concerns/Suggestions:**
1. How is the mHC-style baseline implemented? Is it an exact reproduction of the original DeepSeek paper?
2. The claim regarding interpretability is unconvincing. Since the model is initialized with a bias towards identity, the observed interpretability may be a **direct result of initialization rather than training**.
3. In BE-HC settings, the relation between K (the number of bases) and the expressive power should be studied.
4. Existing studies, such as [1], utilize dynamic weight vectors for all permutation matrices, hence enhancing the expressive power. Since BE-HC relies on the same underlying concept but selects different bases, I suggest further investigation into the trade-off between expressive power and computational/memory complexity to justify the proposed design in **exact settings**.
5. In your 1000-layer training, it is necessary to use mHC as a baseline to **further support your viewpoint** that *"exact is somehow better than approximate (in extreme deep NN training)."* Judging by the results of the 200-layer and 500-layer models, it seems mHC would likely converge at 1000 layers as well. However, the main text and appendix only use ReZero as a baseline, despite the abstract stating that "ReZero and other baselines fail to converge."
6. Similarly, other theorems and experiments are needed to further enhance your viewpoint, instead of merely $(1.00x)^y$ is large.

**Minor:**
1. The demonstration in Figure 1 doesn't align with the content.
2. Some experimental details are missing.

Reference：
1. Yang, Y., & Gao, J. (2026). mHC-lite: You Don't Need 20 Sinkhorn-Knopp Iterations. arXiv preprint arXiv:2601.05732.

---

### Official Review · Reviewer_YAQr · 2026-02-24

**Fit:** 3
**Significance:** 2
**Confidence:** 3

**Summary:**

This paper presents a new approach for constructing doubly stochastic mixing matrices based on the Birkhoff-von Neumann theorem. The approach enjoys favorable theoretical guarantees, and also performs competitively in practice, allowing trainability of ultra-deep networks.

**Strengths:**

The interpretability aspect of permutation matrices used in BE-HC is particularly interesting. I think there lies strong potential therein.

**Suggestions:**

- The paper could use a better introduction to mixing matrices and their place in modern architectural design.
- What's point being made in Proposition 2.4? I don't understand why we care for the mean of the input to remain unchanged. Also, the proposition mentions $\mathbf{y}$, which isn't defined.
- You repeatedly mention the possible ill-effects of Sinkhorn-Knopp, especially because of the finite number of iterations. However, Table 1 suggests that the maximum spectral radius of the mixing matrices is $1$, even for a 500-layer network. I do agree that the accuracy is lower, but it doesn't seem to be because of the spectral radii of the mixing matrices.
- You mention large spectral radius can be harmful, causing gradient amplification, yet unconstrained mixing matrices outperform other methods. This seems to suggest that doing away with sophisticated constructions of mixing matrices may not be harmful.
- I'd be interested in an ablation studying the effect of the hyperparameter $K$, the number of permutation matrices used.

---

### Official Review · Reviewer_wJan · 2026-02-25

**Fit:** 3
**Significance:** 2
**Confidence:** 2

**Summary:**

The authors propose using the Birkhoff-von Neumann theorem to eliminate approximation error in extremely deep neural networks (1000+ layers). Previous work approximates doubly stochastic matrices, leading to accumulated residual error. They hypothesize that exact spectral stability is needed for training at extreme depths.  Using the Birkhoff-von Neumann decomposition to construct exactly doubly stochastic matrices allows for stable training at 1000 layers. The main findings are that approximation error is a bottleneck for extreme depth training, and exact guarantees are needed for training.

**Strengths:**

* Presentation is clear
* The method is efficient and quantization-friendly
* Strong theoretical guarantees
* Clear scientific hypothesis supported through solid experimentation and ablations

**Suggestions:**

Major Suggestions
* The introductory scientific question asks about networks of 1000+ layers, but the visualized results in the main text (Table 1, Figure 1) focus on 200-500 layers. Are 200 and 500-layer networks considered "extreme depths"? Consider rewording the main question so that these results are incorporated.
* To address the significance of the raised question, it would be good to mention the broader benefits and importance of stably training extremely deep neural networks. This motivation can go in the Introduction or Conclusion.

Minor Suggestions
* The style heading is for Principled Design for Trustworthy AI workshop. Change to Sci4DL workshop.
* The legends in Figure 1 cover parts of the graph. Move them to the bottom right corner for (a) and (b) to improve clarity.
* In Figure 1 b, the "X Attention" label is confusing. I recommend removing the "X" or including it in the caption.
* The Figure 1 caption doesn't match the subfigures. It currently says "Extreme depth training dynamics at 200 layers," but panel (b) is 500 layers. It says the (Center) panel shows spectral radius, but the spectral radius plot is actually panel (c). The (Right) panel is labeled "Gradient norm," but this doesn't match subfigure (c). It would also be helpful to refer to the subfigures with (a) (b) (c) in the caption instead of using "left" "right" and "center", which correspond with their subtitles.
* In the Figure 1 caption, include a brief summarization of the key findings for subfigures (a) and (b).
* References for Figures 1-4 should be included in the text for analysis and discussion.
* In the Figure 2 caption, use (a) and (b) rather than (Left) and (Right) to maintain consistency. Include a brief summary of the main finding for panel (a).
* Table 4 in C.5 caption should go above the table in accordance with the other table captions.
* The note on Table 5 should be placed underneath the table so that it remains centered.
Figure 5 should have descriptions in the caption for all 4 panels (a) (b) (c) and (d) rather than just "Top" and "Bottom", which I don't believe cover all the images.
* Include a reference for ReZero in the introduction line 040.

---

### Meta-Review · Area_Chair_ys1V · 2026-03-02

**Recommendation:** Accept

**Metareview:**

I recommend accept based on reviews

---

### Decision · Program_Chairs · 2026-03-02

Accept